# Temporal Dynamic Methods for Bulk RNA-Seq Time Series Data

**DOI:** 10.3390/genes12030352

**Published:** 2021-02-27

**Authors:** Vera-Khlara S. Oh, Robert W. Li

**Affiliations:** 1Animal Genomics and Improvement Laboratory, United States Department of Agriculture, Agricultural Research Service, Beltsville, MD 20705, USA; robert.li@usda.gov; 2Department of Computer Science and Statistics, College of Natural Sciences, Jeju National University, Jeju City 63243, Korea

**Keywords:** RNA-Seq, time series, temporal dynamic methods, differential expression analyses, unsupervised clustering, deep machine learning, meta dynamics, disease progression

## Abstract

Dynamic studies in time course experimental designs and clinical approaches have been widely used by the biomedical community. These applications are particularly relevant in stimuli-response models under environmental conditions, characterization of gradient biological processes in developmental biology, identification of therapeutic effects in clinical trials, disease progressive models, cell-cycle, and circadian periodicity. Despite their feasibility and popularity, sophisticated dynamic methods that are well validated in large-scale comparative studies, in terms of statistical and computational rigor, are less benchmarked, comparing to their static counterparts. To date, a number of novel methods in bulk RNA-Seq data have been developed for the various time-dependent stimuli, circadian rhythms, cell-lineage in differentiation, and disease progression. Here, we comprehensively review a key set of representative dynamic strategies and discuss current issues associated with the detection of dynamically changing genes. We also provide recommendations for future directions for studying non-periodical, periodical time course data, and meta-dynamic datasets.

## 1. Introduction

Owing to rapid advances in sequencing technologies and affordable costs, more complicated experimental designs and clinical applications, such as time course data and meta temporal dynamics, have become feasible and popular in genomic research [1,2,3,4,5,6,7]. Over the past decade, numerous statistical and computational strategies for the characterization of dynamically changing genes over a particular time period have been developed [8,9,10,11,12,13,14,15,16,17,18,19,20,21,22,23,24,25,26,27,28,29,30,31,32,33,34,35,36]. Nevertheless, such dynamic methods have their own pros and cons and are limited in their capability to fully characterize temporal dynamic biological processes [32,37,38,39,40,41,42,43]. In addition, there exist critical and challenging issues that should be addressed in the development of methodological research to detect temporal changes. 

First, the complexity of the architecture in identification of significant dynamic changes that are involved with cellular functions and molecular processes over a series of time points should be carefully examined by a set of coordinated gene structures in an ensemble fashion of multivariate gene-to-gene approaches, as well as single gene-by-gene testing in a univariant strategy. Second, it is well-known that pre-processing, such as normalization procedures, is required for making samples comparable and for adjusting sample-to-sample variations of biological or technical origin in sequencing based temporal dynamic data [44,45,46,47,48,49]. Moreover, it is worth noting that normalized data are often associated with unwanted systematic artifacts due to batch factors, implying that the commonly employed normalization methods do not fully adjust such systematic biases of sample-to-sample variations [1,44,50,51,52,53,54,55]. Third, unlike static methods, many representative dynamic counterparts widely adopted in the community have not been thoroughly evaluated and validated in large-scale comparative studies. It is, therefore, critical to benchmark these methods, particularly to verify their performance and rank multiple dynamic methods in several time course data in terms of power and accuracy of detection of dynamically changing genes [36]. In addition, it seems relevant to evaluate pre-processing procedures inherent in these methods including, but not limited, to pre-filtering, normalization, and batch correction of systematic artifacts [36,47]. 

Lastly, one of the significant advances is the conception of meta-framed data analysis in which multiple time course datasets collected from different laboratories or multiple data generated by different platforms are fully integrated [1,2,3,4,5,6,7]. While these integrated data have made it possible to access a wealth of dynamic data resources, investigators are required to more carefully recognize how to reduce systematic biases/artifacts in data integration in each step of the experimental and analytical pipelines. 

The central goal of this review is to provide well-documented guidance for the analytical pipelines by examining existing dynamic methods in both gene-wise testing strategies and gene-to-gene interaction tools, as there are currently no unanimously validated dynamic methods that are deemed optimal under various scenarios [36]. Further, we attempt to summarize current challenges in dynamic approaches and discusses the importance of integrated multi-platform genomic data in uncovering various cell lineages in differentiation and disease progressive models. 

## 2. Single Gene-by-Gene Testing for Non-Periodical Time Course Data

RNA-Seq has considerably revolutionized the transcriptome studies in the last decade and more complicated experimental designs has been popularly conducted in temporal (and/or spatial) data allowing main biological factors of interest and other nuisance factors at each time point and even in integrated meta-data [1,3,8,16,18,32,39,56,57,58,59,60,61,62,63,64,65,66,67]

In the statistical testing of differential expression analyses, basically static methods [68,69,70,71,72] assume that the collected samples at a fixed time point between treatments/conditions/groups are independent. In contrast, the temporal dynamic methods [1,15,26,62,66,73] precisely account for the time dependent data structure to define time-varying trajectory patterns by assuming that the expression levels at previous time points affect those of later time points and the observed samples between different time points could not be independent, but they are dependently correlated in neighboring time points, such as auto-regressive models, hidden Markov approaches, natural cubic spline functions, non-stationary gaussian processes, and state-transition theory. Importantly, inference of time-varying trajectories is performed in the setting of individual gene-by-gene testing. Supposedly, one gene is not inter-correlated with another gene, or a set of genes; and that each gene behaves independently during biological processes [12,14,15,19,20,21,26,74,75]. The previous dynamic studies for microarrays have contributed significantly and the corresponding literatures can be found elsewhere [76,77,78,79]. Recently, the remarkable advances in developing dynamic gene-wise tools on RNA-Seq time course data have been made. Here we review a set of the representative gene-by-gene dynamic methods for RNA-Seq time course data in this section (also listed in Table 1). 

A. Next maSigpro (prev. maSigPro) [14]: It uses either one- or multi-series time course data as input. It is based on a polynomial regression model for a time variable. A stepwise best fitted model selection procedure identifies genes that are significantly differentially expressed in a temporal manner, based on R-squared values, multiple-corrected p-values, and estimates of coefficients. 

B. DyNB [16]: It has been developed for two series time course data based on non-parametric gaussian process and significant temporal alternations are inferred using a function of time variable in the regression model of gaussian process. Hyperparameters are estimated by the Metropolis Hasting (MH) algorithm in Bayesian sampling technique. Based on negative binomial distribution for read counts, over-dispersed sample variability is assessed. 

C. EBSeq-HMM (prev. EBSeq) [26,80]: It takes into account the sequential property of time course experimental designs in an auto-regressive hidden Markov approach. Gene or isoform level quantification is required as input. It aims at detecting dynamically changing genes (or isoforms) between time points, detected by posterior probabilities, representing each state with a differential expression (DE: Up/down), or equal expression (EE) as the outcome. The time-varying trajectory patterns are further classified into clusters with similar patterns of paths. 

D. ngsp [29]: It is used for either one- or two-series time course data. The statistical rationale relies on a non-stationary gaussian process (nsgp) model using the Bayesian technique [83]. In contrast with discrete inference of temporal dynamics, ngsp estimates various continuous dynamically changing patterns by accounting for the unobserved intervals between given time points across the entire time period. It is advantageous when investigators need to interpolate temporal dynamic patterns beyond such sparsely measured time points due to limited sampling procedures and costs associated with sequencing large-scale time course experimental designs. 

E. lmms [19]: Linear Mixed Model Splines (LMMS), another unified dynamic method, has been developed for multi-factorial time course data obtained from microarrays and proteomics. Notably, this unified strategy has the following functionality: (1) Pre-filtering procedure, (2) identification of individual gene/molecule to represent dynamic changes, and (3) clustering for omics-type time course data. The unified workflow in LMMS between differential expression in multi-factorial time course data and clustering techniques, as well as pre-filtering procedure make it an excellent launch pad for developing improved integrative methodologies for dynamic data.

F. timeSeq in NBMM [12]: It was developed by Negative Binomial Mixed Effects Model (NBMM) with main factors, time, and condition, and an interaction term in the setting of two series time course RNA-Seq data. It allows to distinguish non-parallel versus parallel temporal changes between conditions over time by employing bivariate function of time, which is similar to a two-way ANOVA analysis. 

G. splineTimeR [15]: Originally developed for two series time course data in the microarrays, it is a unified strategy to combine (1) the identification of significantly temporally differentially expressed genes and (2) gene association networks (GANs) relevant to controlling key pathways and biological functions between different conditions over time. 

H. ImpulseDE2 (prev. ImpulseDE) [21,62]: It has been proposed for one- or two-series time course data, particularly for characterizing early on-set perturbated dynamic alterations, such as impulse-like changes. Updated ImpulseDE2 has been released for longitudinally measured RNA-Seq and Chip-Seq data [62]. ImpulseDE2/ImpulseDE is not appropriate for datasets with fewer than six time points. This is a common caveat in practice. However, this method is suitable for moderate or large series of time course RNA-Seq data. The main work-flow is conducted by k-means clustering in the iterative optimization clustering procedure that the pre-user defined k value is not needed. Next, the parameters based on mean-expression profile within each cluster are then fitted to each gene model by minimizing sum of squared errors. The statistical significance for each gene model is given by false discovery rate (FDR). 

I. Trendy [73]: It is developed on the basis of the segmented regression modeling approach with breakpoints for the given time points. The optimal number of breakpoints in the model is chosen by the Bayesian Information Criterion. This method provides the estimated breakpoints to represent statistically significant changes of patterns between time points for up/down/steady expression, corresponding segment slopes, and the adjusted R-squared values for goodness-of-fit. The detected dynamically changing genes are further grouped by co-expression patterns in the visualization procedure implemented by R/Shiny. 

J. AR (auto-regressive model) [1]: It has been developed for single-series longitudinal RNA-Seq time course data by accounting for serial correlations in the auto-regressive model between successive time points using the Bayesian technique. This method provides the posterior probabilities for model parameters. A tail probability is also given representing the statistical significance of serial correlation between neighboring time points for each gene model that can be directly compared to other non-Bayesian methods with multiple-corrected p-values. However, this method is with some limitations: It requires a balanced longitudinal data with equal sample size between time points; additionally, as the number of time points increases, it takes much longer to run than other dynamic methods. The method has been implemented by R and Winbugs and can also be run in JAGS.

K. MAPTest [20]: It relies on maximum average power testing while controlling the average type I error rate [84]. The underlying distributional assumption is given by a K-component latent mixture gaussian negative binomial model in a finite mixture structure. This method has been designed for two series time course RNA-Seq data and is especially suitable for longitudinal settings in repeatedly measured samples. The method facilitates classification between non-parallel differential expression or parallel differential expression as in previously discussed timeSeq [12] of NBMM dynamic method. These two methods have not been directly compared, though. 

L. TIMEOR (Trajectory Inference and Mechanism Exploration using Omics data in R) [85]: It is a user-friendly web-server interface to infer gene regulatory networks using time series multi-omics data: Such as RNA-Seq time series data, Chip-Seq protein-DNA binding sites, and motif information. The goal of this versatile dynamic tool is to provide the complete analytical pipelines to define the interactive gene regulatory networks between genes with the temporal alternation and transcriptional factors. The entire workflow includes (1) pre-processing procedures including alignment, quantification of expression abundances, sample-to-sample normalization, and further correction of batch factors using Combat-Seq and Harman R package; (2) central analyses to detect dynamically changing genes and their regulatory trajectory patterns using existing dynamic gene-wise tools and clustering R packages, ImpluseDE2, next maSigPro, DESeq2, Cluster, and ClusteProfiler; and (3) further investigation for potential coherent gene functional activities based on gene ontology and enrichment pathway analysis using existing pathview and STRING R packages, and de novo motif finding step with MEME, respectively. 

M. TimeMeter [64]: Another R package, named with TimeMeter has been proposed to facilitate the comparative time series of gene expression profiling data in terms of time-shift patterns with similarity on temporal dynamics or differentially progressing patterns of dynamically changing genes (with different speed of dynamic changes). In a proof-of-principle, it is based on dynamic time warping algorithm to align two different sequences, i.e., query versus reference with various temporal patterns by further incorporating the advanced features, (1) truncating the sequences in order to make the time windows comparable each other (2) computing the metrics of similarity patterns: Percentage of alignments; spearman rank correlation between two alignments; and statistical significance, and (3) progression advance score based on the slopes in segmented piecewise regressions to represent how much deviated from the diagonal line. Thus, this dynamic method is designed for the comparison of two different set of single series of time course data. 

N. PairGP [61]: This dynamic method has been implemented by python to account for the longitudinally measured time series data with multiple conditions, especially when having the same biological replicates to be measured in different conditions, i.e., paired multi-group conditions. The rationale of this method is based on a non-stationary Gaussian process with the exponentiated quadratic kernel functions for the response model for treatments, pairing effect model within the same biological replicate, and random fluctuation noise. Given the number of conditions, the log marginal likelihood for each partitioning set from Bell number is evaluated with the base response model that does not have the pairing effect. This dynamic method is designed for a multi-series of paired longitudinal time course data in which the same biological replicates are matched in different conditions.

O. GPrank [81]: This dynamic method has been implemented by R package and the rationale of this method is on the basis of two different Gaussian process models, time-dependent versus independent model using the estimated mean abundance matrices and corresponding variances for a given single series of time course data. Time dependency is captured by squared exponential, radial basis kernel function and the temporal changes of gene expression patterns are evaluated by the natural logarithm of Bayes Factor to rank the most significant dynamically changing genes (or other genomic elements). This method is robust for a short and irregularly measured time points in single-(one-) series of time course data.

P. dream (differential expression for repeated measures) [60]: This dynamic method has been implemented within a variance Partition Bioconductor package, incorporated with limma and voom. And this method has been applied for the large-scale of cohort studies [63,86] with multiple condition groups (e.g., four different brain regions) in a multi-series of longitudinally-measured time course RNA-Seq data. This method is based on linear mixed models to account for arbitrary multiple random effects for a given particular gene, varying variances terms across genes, precision weight functions considering random effects for samples within-individual, and small sample issues for hypothetical testing with Kenward–Roger approximation. 

Q. rmRNAseq [82]: As another voom-incorporated R package, it has been also proposed for the longitudinally measured multi-series of time course data to account for correlated samples within-individual. It is based on generalized linear model with the continuous autoregressive correlation structure, parametric bootstrap method for estimation of temporally differential expression, residual maximum likelihood for estimation of parameters.

State-Transition Analysis by Rockne et al (2020) [87] has been conducted to identify the critical points representing the initiation and development of acute myeloid leukemia disease progression from the reference normal hematopoiesis to disease state of leukemia. It is mathematically based on state-transition theory, stochastic differential equation for a double-well quasipotential of 4th-degree polynomial in an energy function with "w"-shape of two stable valleys and an unstable peak, and eigengenes in principle component analyses. It defines the critical points during the development of leukemia in a given time points, (1) stable reference normal state of hematopoiesis in control group, (2) stable reference state of perturbed hematopoiesis without evidence of disease, (3) unstable transition state from normal to leukemia, and (4) stable state of leukemia. This method has been analyzed for two-series of time course RNA-seq data. However, source codes and manual for this dynamic method with an example data are not currently available in public. Albeit ChromTime is designed particularly for Chip-Seq time-series data [88], we wanted to mention ChromTime for Chip-Seq time course epigenomics data here, as we will discuss the integrative dynamic tools in later sections, including Chip-Seq data. It has been developed for modeling spatio-temporal changes of chromatin marks as the dynamic peak caller. The main conceptual idea fully accounts for the inference of the territorial peak boundaries in blocks in genomic sequences and then defines their different behaviors in spatial patterns, expanding, contradicting, and steady during time points. The estimation of optimal model parameters is given by expectation maximization. Thus, the advantageous feature of this spatio-temporal peak caller for the dynamic epigenomic data sets, such as Chip-Seq and other variants enables to better characterize the temporally differentially changing peaks and the changes of spatial structures in territorial boundaries of peaks including asymmetric patterns and longer peaks.

Due to the lack of holistic comparative studies of various dynamic methods, there will be no universally or widely accepted or best methods available for different scenarios under time course experimental designs. Therefore, we recommend the development of a complete analytical pipeline to better characterize dynamic changes and to reduce misleading results (Figure 1) with detailed descriptions for each step) [1,6,10,15,17,19,20,23,25,28,33,44,46,47,49,50,51,53,54,55,58,59,62,67,68,71,72,74,75,76,81,89,90,91,92,93,94,95,96,97,98,99,100,101,102,103]. 

## 3. Single Gene-By-Gene Testing for Periodical Time Course Data

Circadian rhythms can regulate periodical gene expression according to daily or 24-h (or 8, or 12 h) oscillations. Over the last decades, the genetic regulatory mechanisms of circadian genes have been thoroughly explored to characterize clock-controlled dependency, such as that observed in physiology, metabolism, and mental illness [10,13,17,22,27,30,31,33,35,41,42,43,74,75,104,105,106,107,108,109,110]. Moreover, it is evident that aberrant expression patterns and mal-functionalities in circadian clock-controlled genes, as well as their oscillating systems, have been highly associated with human diseases and therapeutic effects in the field of biomedical research and pharmaceutical chronotherapy [110]. Here we review gene-wise methods for circadian rhythmic changes and cell-cycling genes in RNA-Seq periodical datasets (Also listed in Table 2). 

A. JTK_CYCLE [74]: It was originally developed for microarrays to identify a certain set of genes representing periodicity of circadian rhythms in terms of period length, phase, and amplitude. The main rationale is based on a non-parametric strategy that includes the Jonckheere–Terpstra method, which characterizes the monotonic patterns in ordered groups over time, and Kendal’s tau rank correlation between two groups. The JTK_CYCLE statistic is given by the exact permutation null distribution, resulting in multiple corrected p-values for each gene. Later, Hutchison et al. (2014) [100] further improved JTK_CYCLE with empirical p-values by comparing other competing methods including the original JTK_CYCLE, ANOVA, Fourier projection method based on microarray periodical data sets.

B. MetaCycle [17]: It is the result of the integration of multiple existing methods for periodical time course data, ARSER (ARS) [35], JTK_CYCLE [74], and Lomb–Scargle (LS) [30]. Fisher’s combined *p*-value is used to define a common set of candidate genes among three dynamic methods. In two built-in functions within this meta-method, the meta2d function to integrate outcomes from three different dynamic methods, whereas the meta3d function is to merge outcomes from multiple individuals (biological replicates) within a single dataset by choosing a specific dynamic method, such as JTK_CYCLE, or ARS or LS. Thus, this method can be extended by combining additional methods to identify periodicity in the later version of the meta2d function. However, the current version does not allow to integrate results from three dynamic methods and multiple individual sets simultaneously. 

C. RAIN (Rhythmicity Analysis Incorporating Non-parametric method) [102]: It has been developed to detect periodical changes, amplitudes, phases, and peaks to better account for asymmetric behaviors such as steep rises and slow fallings, and vice versa in umbrella alternatives, especially compared to JTK_CYCLE test. This robust method has been implemented in R package and a user-friendly web-server interface.

D. DODR [103]: Unlike the aforementioned tools, it has been proposed to identify differential rhythmicity in the factorial time course designs in periodicity in which there are two different condition groups such as mutant vs wildtype samples in animals and humans. The rationale is based on both parametric and non-parametric rank-based tests by assuming non-Gaussian measurement noisy errors to detect significant alterations in rhythms, amplitudes, and phases. 

E. LimoRhyde [101]: When compared to DODR and other naive periodical methods, it is a more generalized strategy to allow more complicated multi-factorial time course designs in periodicity where two or more conditions at each time point for external stimuli and other covariates such as age, gender, and other batches are available. Basically, it is designed to identify a certain set of genes with differential rhythmicity indicating the significant changes both between conditions and during the zeitgeber/circadian time period by using cosinor regression decomposing zeitgeber/circadian time onto sine or cosine period of 24 h and empirical Bayes approach for shrinkage sharing information across genes analogous to limma. 

## 4. State-Of-The-Art Batch Detection Methods for Removing Unwanted Biases in Data Integration 

The batch issue in data integrations not only an RNA-Seq specific problem inherent in the current technology, but also an issue that has been steadily discussed in the history of high-throughput of datasets [1,44,46,48,50,51,52,53,54,55,58,91,111,112,113,114]. Nevertheless, most existing dynamic methods solely focused on the inference of dynamic trajectories, frequently failing to capture the issues related to precise prerequisite analytical procedures that can reduce noise or bias. Here we highlight several advanced batch detection methods that are able to characterize and remove unwanted systematic artifacts due to nuisance factors. (Also listed in Table 3).

A. ARSyN (ASCA Removal of Systematic Noise) [113]: It has been originally developed for time series microarrays by using ANOVA simultaneous component analysis [115], wherein it is primarily utilized as a simple exploratory diagnostic tool rather than the batch correction method. As the extension of the initial approach, the residuals of raw expression profile data are identified, subtracted from the terms, noise of signals, and signals of noise from data decomposition. This ARSyN batch tool provides the adjusted expression profiling data after batch removal that can be directly utilized in the subsequent step to infer time-varying trajectories. The major advantage is the ability to identify both known and unknown batch factors. Prefiltered and normalized data can be used as input. 

B. Combat-Seq/Combat [55]: It is based on by empirical Bayes (EB) estimates for batch effects by borrowing information common across genes when removing the batch effect in terms of location and scale parameter from the standardized expression data for each gene, which could be more robust estimates when small sample size is available for given batch factor. The latest version of Combat (Combat-Seq) has also been developed for RNA-Seq data by accounting for the count property in the identical scale of log link function in generalized linear modeling (GLM) approaches when there are known batch factors. As ARSyN, it also provides the batch-free expression profiles that can be directly applied for dynamic gene-wise methods. For input data, the pre-filtered and normalized data can be used. 

C. RUVSeq [46]: Similar to Combat-Seq/Combat, it is specifically designed for RNA-Seq data. As the input data, normalized counts by edgeR, DESeq2, or upper-quantile normalization can be uploaded. It has been developed from factor analysis of singular value decomposition (SVD). Depending on the use of negative control genes/samples as the reference set representing the constant expression patterns for a biological factor of interest, RUVg/RUVs is employed. When such negative reference sets are unavailable, residual expression profiling matrix subtracted from the fitted values of the primary variable, such as tissue type, can be applied for RUVr. This tool can be applied for both cases when batch factors are known and unknown and it also allows the multiple batch factors. However, users should select the number of hidden factors (k value) for unwanted batch sources. RUVSeq batch tool is used for simple pairwise or multi-group comparison at a fixed time point in RNA-Seq static data. However, it is unclear whether or not the incorporation of estimates for batch factors on each of the trajectory models is used to identify temporally differential expression of dynamically altered genes. 

D. svaseq/sva [52,91]: It is based on the following steps: (1) Extracting the residual expression profiling data, subtracted from the fitted values of the primarily main factor of interest in the study; (2) based on the residual expression data, the selection of orthogonal eigenvalues on singular value decomposition indicating surrogate variables; and (3) the statistical significance of each eigengene under the null bootstrapping distribution. The detected batch factors can be incorporated with a subsequent differential expression method, such as limma; however, this batch correction method does not provide a batch-corrected expression matrix. Recently, it has been further developed in the updated version of the original sva with svaseq, and logarithmic transformation is carried out on the same scale as the GLM model by considering the count property for RNA-Seq datasets. Similar to RUVg, a set of negative control genes that could be affected by batch factors, but not affected by primary biological factors of interest, are used to infer batch effects, known as supervised svaseq (ssvaseq). However, such inferred batch factors cannot be directly applied to each of the dynamic methods that have been developed for single-series (including longitudinal data), case-control time course, multi-series time course, or periodical data sets.

E. gPCA (guided-PCA) [90]: It is also based on the extension of singular value decomposition in the naive principal component analysis (PCA) method using a batch indicator matrix to represent whether the sample belongs to the batch factor or not to replace the original normalized expression matrix (referred to as the guided-PCA in the study). The method was developed using the score metric to measure the proportion of the total variance contributed by the batch factor as the ratio by comparing the explained variances between guided-PCA vs naive PCA method. The statistical significance is evaluated using the permutation test by randomly reshuffling the labels of samples in the batch source. However, this method is not applicable when the batch factor is unknown and where exist multiple batch sources in the complicated dynamic datasets, such as dynamic meta datasets. 

F. Harman [58]: Analogous to Combat-Seq/Combat and ARSyN, it also provides the batch-free expression profile data after the removal of unwanted systematic biases, which can be directly applied for any types of time course dynamic tools. As already discussed in the previous studies [46,51,52,113], note that it is very critical to preserve the true biological signal when removing the batch noise. Thus, they proposed another PCA-based batch correction method while well preserving the true biological signals and removing batch noise in the simple algebra form and its performance has been compared to Combat and uncorrected data. This method is also unsuitable to the datasets in which the batch is unknown to investigators or there exist multiple batch factors.

G. Maximum Mean Discrepancy and Residual Nets (MMD-ResNet) [89]: It is based on the mathematical properties of Maximum Mean Discrepancy, loss function, Residual Neural Network strategy for more recent technologies, protein mass cytometry and single-cell RNA-Seq data. In principle, adjusted (calibrated) sample data after removal of the batch effect are generated by residual neural network techniques. In its current version, the input data are analyzed with only one batch factor, regardless of the type of platforms for various modalities and data types, either static or dynamic time course data. This method cannot be run for more complicated cases, when there exist multiple batch factors and unbalanced samples within a batch factor. the improvement can be made to address these issues in. The capability to use in well-balanced experimental designs are always recommended.

In conclusion, the use of Combat-Seq/Combat, gPCA, and Harman, is generally limited because batch factors are identifiable and known to investigators only in few case studies. RUVSeq works for studies with both known and unknown factors; however, users need to select the k value as an arbitrary number. svaseq can also be applied for unknown batch cases. Both methods have been originally implemented for RNA-Seq specific static multi-group studies. As for dynamic time course datasets, we recommend Combat-Seq, svaseq, and RUVSeq as default batch detection methods, which can be performed with pre-filtered and normalized datasets together in the initial step of exploratory diagnostic analyses. However, to incorporate detected batch effects with subsequent dynamic methods to identify time-variant trajectory patterns in RNA-Seq time course datasets, Combat-Seq and Harman should be used since both provide adjusted expression profiling data after batch removal when batch factors are known. Additionally, MMD-ResNet can be applied for dynamic specific designs with single batch factor as it is not restricted to any specific experimental design and/or platform. 

Batch-free data should be analyzed using dynamic methods to infer time-variance trajectory patterns. Versatile batch detection methods are needed to cover a wide range of dynamic data including non-periodical and periodical data. As meta-dynamic data in the post-genomic era become increasingly available, sophisticated analytical methods are urgently needed.

## 5. Coherent Gene-To-Gene Strategies for (Non)-Periodical Time Course Data 

Genes that are relevant to cellular perturbations by external environmental factors, such as drug treatments, are regulated with several other genes by interacting with trans-acting and cis-regulatory elements in the genetic transcriptional regulatory machinery to represent different biological pathways or differential networks. It is very important to infer more reliable sets of pre-defined genes that are associated with functional pathways, network modules, and clusters of co-expressed patterns in the sparse and irregular time series RNA-Seq data. Motivated by single gene-by-gene dynamic methods, researchers have started to recognize the importance to characterize dynamic gene-to-gene interactions in the following paradigms [8,11,23,28,116,117,118,119,120], unsupervised clustering techniques, gene set analyses, and machine learning strategies including deep neural network for time course RNA-Seq data as well as their integration with other types of omics data sets. 

Basically, the gene list selected from temporally differential expression analyses is further analyzed in subsequent down-stream analyses to identify gene interactions in the separate analytical pipelines [14,19,21,26], such as gene clustering, enrichment gene set test, and gene regulatory networks (GRNs). However, the vast majority of commonly employed gene clustering, gene set tests, and network tools have been originally implemented for static data. Therefore, they provide identical results even when switching sample labels in a time course dataset, which does not practically translate to biological dynamic processes [121,122,123]. Thus, to fully characterize a given dynamic biological function, gene clustering, gene set tests, and differential network modeling approaches should be continuously developed specifically for dynamic data [8,11,23,73,76,98,116,117,121,122]. We review the existing dynamic gene-to-gene interaction tools in this section (also listed in Table 4).

### 5.1. Dynamic Gene Set Analysis Tools

Variance component score test (tcgsaseq, Gene Set Test) [8]: The initial pioneering work [11] has been attempted to identify a pre-defined gene sets that are temporally differentially co-expressed and are also functionally enriched for single-series of longitudinal time course array data, a.k.a. TcGSA (Time-Course Gene Set Analysis) in the R package. It aims to precisely account for the heterogeneity within gene sets due to patients by assigning as the random factor in the linear mixed effects model. For RNA-Seq specific longitudinal data, as the new tool, tcgsaseq, followed by TcGSA has been further developed to identify significant enrichment gene sets. In principle, the variance component strategy has been applied under the assumption of asymptotic distribution by fully addressing mean-variance relationships and small sample sizes within gene sets in a permutation-based test. The robustness of this method has been demonstrated, particularly when samples (subjects) show substantial heteroscedasticity in various longitudinal settings. The input requires log-transformed normalized data. This method should be further updated to run multi-series time course datasets with two or more distinct conditions at each time point. 

### 5.2. Dynamic Clustering Tools

A. funPat (Functional-based Pattern analysis) [23]: It is for both single- or multi-series time course RNA-Seq data. Analogous to splineTimeR and lmms, this approach is also a unified strategy based on the smooth integration of following three main steps. It includes (1) the best fitted gene model for which dynamic gene-by-gene testing under the empirical null hypothesis is selected by the precision values of estimated parameters among gamma, log-normal, and Weibull distributions; (2) funPat linear model-based clustering for time-dependent data structure; and (3) mapping procedures between detected co-expression temporal patterns and the given annotation information of gene ontology and pathways to assess whether or not the significant dynamical changing genes and their temporal trajectory patterns detected in the previous steps are highly associated with the given gene sets.

B. DPGP [116]: It is for single-series time course RNA-Seq data. In order to better incorporate the data-driven features, i.e., (1) time-dependent structure in the inference of time-varying trajectories and (2) the choice of proper number of clusters, this tool is based on the combination of Dirichlet Prior mixture models of Gaussian Processes by estimating the posterior probabilities of model parameters via Monte Carlo Markov Chain simulations. 

C. LPWC [117]: Similar to DPGP, it is also able to handle single series time course RNA-Seq data, such as those obtained from impulse-like perturbation experiments. This method is based on the Lag-Penalized Weighted Correlation (LPWC) approach to fully take into account the lagged temporal profiles between genes by assigning a Gaussian kernel penalty score to reduce the chance of higher weighted correlations for such genes. However, one of the constraints in this method similar to other state-of-the-art-clustering tools, one gene/sample should belong to a cluster. 

EPIG-Seq clustering method [124] allows to include multiple group conditions, although it has been implemented for static data. It incorporates the correlation metric for counts, Wilcoxon rank-sum non-parametric test for detection of magnitude of change between samples, and estimation of dispersion parameter by quasi-Poisson regression model. Trendy tool [73], as discussed before, as a dynamic clustering tool, can also be applied for a single-series time course RNA-Seq. Several research groups have also highlighted the importance of development of dynamic clustering tools for time course experimental designs in conventional microarrays [98,121,122,123,125]. The fundamental structure of time course data contains three-dimensional formats such as genes-conditions-times. While the majority of clustering tools assume that one gene/sample should be allocated to a cluster, as more flexible and improved strategies for unsupervised clustering techniques [65,126,127,128], for instance, Biclustering (a two-way in genes-conditions) and Triclustering (a three-way, e.g., genes–times–conditions and drugs–genes–dose levels) methods have been proposed to better account for the property that a gene/condition/sample/time could be assigned to multiple clusters. This extension to better characterize the higher order of factors/variable sets in time course data structures is critical for properly grouping genes/samples in terms of biological relevance as some of genes are coordinately regulated in multiple cellular processes and biological functions. For a fully comprehensive review and discussion, readers of interest should consult with Hendriquez and Madeira (2018) [127], in which various Triclustering algorithms and their applications in biological time course data and other types of three-dimensional are further discussed.

### 5.3. Dynamic Machine Learning Tools

In order to infer the functional interplay amongst genes affected by the initial stimuli in perturbated cells or other types of time course data, the plethora of dynamic gene regulatory network (dynamic GRN) methods have been steadily proposed from arrays until present Seq-based data. In essence, dynamic GRN methods are mainly grouped into three categories: (1) Continuous state space models, (such as Dynamic Bayesian Network: DBN, Markov Model: MM, State Space Model: SSM, and Ordinary Differential Equation: ODE); versus (2) discrete state space models (such as Boolean Network: BN and Probability Bayesian Network: PBN). And both models account for structure and temporal dynamics when inferring the causal network modules, whereas, there exists (3) Relevant Network (RN) and Bayesian Network (BN) to consider only structure [129]. In this section, we discuss the applications of dynamic GRN methods and deep machine learning tools.

(i) Dynamic GRN tools for RNA-Seq data: dynGENIE3 tool [118] has been developed by using ordinary differential equation and random forest tree-based approaches to define the network modules for both the steady-state and dynamic time course RNA-Seq data. More recently, as another ordinary differential equation GRN method, BINGO tool [119] to capture the interactive gene networks for RNA-Seq time course data has been proposed on the basis of non-parametric gaussian process with Monte Carlo Markov Chain sampling procedures and non-linear function to the probabilities of trajectories in ordinary differential equation instead to the derivatives. It is also based on benchmarked data by comparing some existing tools such as dynGENIE3 [118]. As one of the most recent advances, BETS (Bootstrap Elastic net regression Time-Series) tool [120] has been developed for RNA-Seq time series data with perturbation. For a given time lag, a vector autoregressive modeling approach as the granger causality has been applied for the set of temporally differentially expressed genes (~2800 genes) during 12 h after A549 cells exposed to glucocorticoids. 

(ii) GRNs in traditional microarrays: Time-varying multivariate state space model (SSM) tool [99] is designed for a short series of time course expression profiles where measurements from samples are collected between sparse and uneven time points. With the combination of Hidden Makov model and Dynamic Bayesian Network, HMDBN tool [95] is proposed by incorporating the non-stationary DBNs in which the structure and parameters are not fixed over a series of time points. Accordingly, in the method, Structural Expectation Maximization (SEM) and improved Bayesian Information Criterion sharing information between times are employed. Boolean Function Network tool [24,25] for array cell cycling data has been implemented in Matlab, which requires a transformed expression as input, with a range of (0, 1) interval by an empirically cumulative distribution function. The pre-defined regulators of transcriptional factors should also be uploaded. Note that Boolean Function Network is based on pairwise dependencies between genes to infer the optimal function of global state trajectory over time points. It is fairly a gene-wise approach to identify causal relationship between genes to account for time-delay, however, we have placed this tool in this section of coherent gene-to-gene strategies as it is not a statistical testing procedure for each gene. Additionally, at least ten time points should be included in the input dataset to identify reliable gene regulatory networks (GRNs) that are relevant to pathways and biological functions in cell cycles and oscillations. Time-Delay ARACNE tool [10] has been implemented for identifying gene regulatory association networks on the basis of an information-theoretic approach of mutual information, and statistical dependency to assign gene-to-gene interactions [130]. TD-ARACNE defines GRNs in time course experimental designs, such as cell cycles, by precisely incorporating time-dependent data structure assuming that gene expression level at the current state is affected by the previous time points of other genes. 

Importantly, the inference of dynamic gene regulatory networks based solely on time course RNA-Seq data has limitations as the significant changes on gene expression levels over time points for the particular biological dynamic process are involved with TF, histone modifications (HM), other cis- and trans-acting elements in the gene regulatory machinery. Therefore, we review the GRN methods in time series RNA-seq data and its integration in the following. 

(iii) More informative dynamic GRN tools for data integration: CRNET tool [131] has been proposed to infer functional regulatory networks by incorporating the efficient Gibbs sampling procedures to estimate the hidden TF activities and the posterior probabilities for binding events for the integrated Chip-Seq data and time series of RNA-Seq data. On the other hand, iDREM tool [94] was developed aiming at visualizing the interactive relationships of dynamic GRNs for integration of multi-omics data, including time-series data of mRNA, miRNA-Seq, epigenomics, proteomics, scRNA-Seq, and static TF-gene, and protein–protein interaction data. More recently, a comprehensive review [96] to cover dynamic functional regulatory network tools for time course multi-omics data including different layers, transcriptome, genomics, epigenomics, metabolomics, variomics, and proteomics has been published. For the dynamic specific tools of gene functional regulatory networks, several unsupervised/supervised methods are well described in the review. In addition, the importance to infer more precise regulatory networks for the interactive relationships between platforms from multiple omics data in the system biological aspects of personalized medicine has been emphasized. 

(iv) GRN tools for scRNA-Seq data: While we primarily focus on the dynamic tools for bulk RNA-Seq transcriptome time course data, the recent advances on machine learning tools and their applications for scRNA-Seq studies have been omitted for the sake of brevity. Instead, an excellent comprehensive review [132] for the diverse gene regulatory networks tools using scRNA-Seq data alone, integration with genomics or single cell epigenomics data can be found in the reference. Additionally, a review for GRN methods utilized in single cell RNA-Seq data is also available [133]. 

(v) Dynamic differential networks: In order to define differential networks representing group-specific differences over time using inferred gene regulatory networks, DryNetMC tool [28] has been developed. Input file formats include, normalized read counts, such as those normalized by DESeq or further adjusted by advanced normalization methods [49,68,71,72,92] in case-control time course RNA-Seq data. Thus, this dynamic method is not suitable for one- or multi-series time course data with more than two levels of the condition group. Furthermore, parameters to infer sample-to-sample variations for biological or technical replicates in the model are not defined. 

As a differential network tool for static multi-omics data (miRNA, mRNA, and proteomics), iDINGO tool [134,135] has been proposed to define group difference on differential networks (e.g., disease patient groups vs normal controls). Differential networks are defined by the conditional dependency by simultaneously accounting for both intra- (within platform) and inter- (between platforms) conditional dependencies. The input files can include up to three different matrices with matched samples among platforms while the output generates the multiple correction p-value across edges in differential networks and visualization outcomes that show group-specific changes in the chain graphical model. As the method allows only two group conditions at each platform, the current version is not applicable for multi-group conditions or continuous types of covariates, such as age and time. 

(vi) Deep learning neural network approaches in current genomics data: We also want to describe the deep machine learning tools that have been widely applied to current genomic data analyses for the purpose of predictions and classification problems. A comprehensive review to fully discuss their applications in the epigenomic and genomic data is currently available [136]. As discussed in the study [136], while the ability to interpret and explain for the selected features from neural network methods is still needed to be addressed, current approaches for interpretation methods can be partitioned into two major categories: 1) Convolution neural network (CNN), sub-categorized into input modification, deconvolution, and input reconstruction and 2) recurrent neural network (RNN) in attention mechanism based on the remarks of Grün et al [137] and Singh et al [138]. These deep learning tools can be used for the identification of (1) motif discovery for DNA/RNA sequence alternations associated with protein binding, (2) epigenetic effects on DNA sequence alternations, (3) chromatin interactions and predictions, (4) gene expression prediction, and (5) non-coding RNA identification and regulation [136]. Further, a prediction deep learning tool [139] for gene expression has been proposed by given information of knockout experiments and master regulator genes in perturbations, compared to others including classical RNN and bidirectional RNN. On the other hand, a prediction tool KEGRU [140] for transcription factor binding sites (TFBSs) has been developed by the means of bidirectional gated recurrent unit (GRU) networking and k-mer embedding, which are utilized for identifying the feature information from DNA sequences. The feature information is then used for the prediction of TFBSs. As dynamic Bayesian neural networks, hidden Markov neural networks can be applied for time series data [97]. The detailed mathematical formulas, notations, and algorithms have been given by [97], though they discuss the applications on non-biological time series data sets. 

## 6. Meta Analyses in Cell-Lineage Differentiation and Disease Progressive Models 

More recently, several studies [4,6,7,16,32,66,67,141,142] of high-resolution genome-wide datasets obtained using different platforms have been conducted to unravel molecular mechanisms underlying cell differentiation using various models, such as in T17 cells, hematopoiesis of blood formation, mouse bone-marrow induced dendritic cells to stimulation, and dynamic response of yeasts to lipopolysaccharide stimulation. Integration approaches between RNA-Seq and Chip-Seq/Methylation/Proteomics/microRNA have been utilized in developmental biology, stimuli-response animal models, pharmacogenomics to characterize target therapeutic effects on treatments, disease progressive models for tumors, as well as in age-related human diseases, such as Alzheimer’s disease [4,5,67,143,144,145,146,147,148]. These studies [141,142] highlighted the importance of characterization of chromatin state dynamics during cell differentiations. Moreover, mechanistic understanding of perturbations during differentiation, using synergistically integrated strategies between Chip-Seq and RNA-Seq, in time course experimental designs will allow us to dissect biological pathways controlling these processes. Large-scale of meta-analyses across different platforms have been conducted to increase the power of detection and to fully characterize dynamically changing genes and their functional roles that are related to explicitly or implicitly causal and consequential effects in the complexity of trans-acting and cis-regulatory elements that regulate biological processes. 

Altogether, it is timely crucial to establish general framework and guiding principles for analyzing various time course datasets with well validated dynamic methods that will reduce method-, study-, and platform-specific artifacts, which will lead to a conclusive consensus and reproducible results in the diverse range of dynamic studies, such as perturbative cellular models, cell-lineage programs, disease progressions, and other types of dynamic processes [7,149]. 

## 7. Summary and Future Perspectives 

Although many of dynamic tools dealing with the diverse types of RNA-Seq time course data in differential expression analyses have been widely adopted for the purpose of identification of dynamically changing genes, more challenges still exist and need to be to be properly addressed. Each method has its own unique characteristics and data distributional assumptions as well as input/output formats, running procedures, and estimated parameters. As shown in static methods without regards to time, such as edgeR and DESeq2 [68,69,70,71,72,92] under the major assumption that observed samples are collected independently at a fixed time point, all dynamic methods with regards to a series of time points should have potential to serve as universal tools handling all necessary pre-processing procedures within their own built-in structure. 

In addition, the comparative study to evaluate the performance between dynamic gene-wise testing methods has been conducted recently [36] whereas there are no large-scale of comparative studies available for the exploitation of the impact of pre-processing steps, the choice of pre-processing tools, or dynamic methods with regards to the inference of dynamically changing genes based on gold-standard reference sets. Such comparative studies should also include various time points (short 3–4 points; moderate 5–7 points; long series ≥8 time points), different numbers of replicates, multiple conditions, and variable noise levels. 

Normalization procedures outweigh differential methods themselves in determine the outcome in the identification of differentially expressed gene sets [36,45,47]. More importantly, significant batch effects, even after normalization procedures, exist due to confounding nuisance factors. Generally, metadata are subjected to higher chances of batch contamination. Unbalanced samples within a batch factor should also be considered in batch detection methods. To better define dynamically changing genes with conditions over time, the issues in pre-processing procedures prior to trajectory inferences should be precisely addressed.

In addition, the development and application of dynamic tools have significant impacts on post-genomic era data analyses, particularly for the characterization of gene-to-gene interactions using unsupervised clustering techniques, pre-defined gene sets, gene regulatory networks, and deep machine learning tools. It is conceivable that the evaluation and validation based on benchmarked datasets with robustness and reproducibility will undoubtedly facilitate our understanding of complex biological processes underlying diseases. Importantly, user-friendly web-based interfaces or packages (instead of separate stepwise analyses) for enhanced and unified analytical strategies (meta-pipelines) that are more capable of characterizing dynamically changing genes, GANs, and their corresponding roles in biological pathways are urgently needed. Integrated dynamic methods should be implemented by addressing the following issues related to time course datasets and their metadata: (1) Normalization, (2) filtering genes or samples with low expression and quality, (3) unbalanced or unevenly measured time points and replicates, (4) batch removal or incorporation of batch factor in the detection model of dynamic changes in the inference of trajectory patterns, (5) dynamic specific methods in co-expression clustering methods, gene set enrichment tests, GRNs, and (6) differential networks to represent time- and condition-specific dynamic changes for personalized medicine. We believe this review serves as a template for more precisely analyzing dynamically changing genes over a broad range of time points or repeated measures in experimental and clinical applications while addressing the necessary next step in meta-dynamic data analysis. 

## Figures and Tables

**Figure 1 genes-12-00352-f001:**
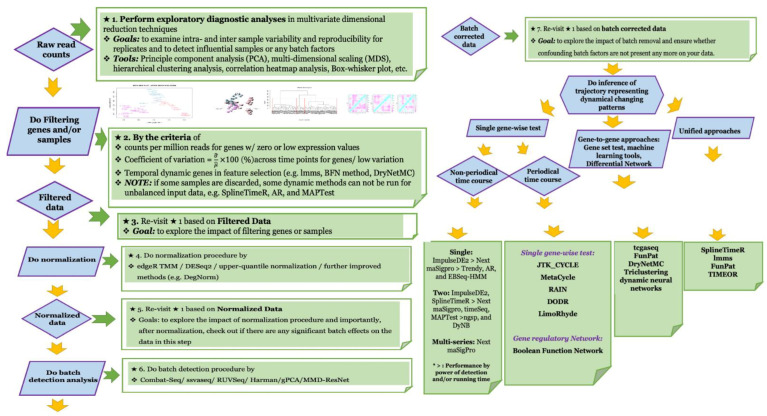
It depicts the schematic illustration of complete analytical pipeline for the dynamic time course data.

**Table 1 genes-12-00352-t001:** Dynamic gene-by-gene (gene-wise) testing tools for non-periodical time course RNA-Seq data.

Tools	Features/Functionalities	# of Conditions Per Time	Experimental Designs (Data Type)	Parameters Used in the Study: T(time), R(rep), C(cond)	Compared Tools in the Study
Next maSigpro [14]	time-based polynomial regression; step-wise best fitted model selection; differential splicing events at isoform levels	w/o condition or w/ two or more	crorss-sectional time course, i.e., non-longitudinally measured samples in single-(one-), two-, and multi-series of time course; balanced/unbalanced time course; normalized data for input	T(4,6), R(2,3,5), C(2)	maSigPro-LM, edgeR
DyNB [16]	non-parametric gaussian process with metropolis hasting;	w/ two conditions	longitudinally measured two-series time course; balanced design	T(5), R(3), C(2)	DESeq
EBSeq-HMM [26,80]	auto-regressive hidden Markov approach; estimates of hidden paths DE (up/down) and EE	w/o condition	longitudinally measured single-series time course; balanced/unbalanced time course at least two replicates per time	T(5,7), R(3), C(NA)	EBSeq, DESeq2, edgeR, voom, maSigPro
Ngsp [29]	non-stationary gaussian process in Bayesian	w/ two conditions	longitudinally measured two-series time course; qPCR time course	T((9), R(3),C(2)	GP two sample
Lmms [19]	linear mixed model splines; unified strategy of pre-filtering, gene-wise testing, and static clustering	w/o condition or w/ two conditions	one or two-series time course; microarray and proteomics	T(4,6,), R(2,5,),C(2)	limma for array
timeSeq [12]	negative binomial mixed effects model with time, condition, and interaction terms; non-parallel vs parallel temporal patterns	w/ two conditions	two-series time course; balanced/unbalanced time course; not handling the variability for replicates	T(6,9), R(1,3), C(2)	MLL-ratio
splineTimeR [15]	natural cubic spline regression; unified strategy between gene-wise testing and gene association network	w/ two conditions	two-series time course; balanced design; replicates are not required	T(7), R(1), C(2)	BETR for array
ImpluseDE2 [21,62]	iterative optimization clustering; the parameters of initial peak and steady state, temporal transitions, and slopes for transitions based on the mean expression profile within each cluster	w/o condition or w/ two conditions	longitudinally measured single- or two series time course, e.g., early on-set perturbated dynamic alterations compared to control group; RNA-Seq and Chip-Seq dynamics	T(6,7,10,23), R(3), C(1,2)	DESeq2, DESea2splines, edgeR, limma, ImpulseDE
Trendy [73]	segmented regression model w/ breakpoint in Bayesian information criterion; the estimates of breakpoints for DE (up/down) vs EE (steady)	w/o condition	longitudinally measured sing-series time course; replicates are not required; microarray and RNA-Seq dynamics	T(17, 25, 48, 50), R(3), C(NA)	EB-Seq, funPat
AR [1]	auto-regressive model based on MCMC	w/o condition	longitudinally measured single-series time course; balanced design; replicates are not required	T(2,5), R(8), C(NA)	Next maSigPro-GLM, DESEq2, edgeR
MAPTest [20]	maximum average power testings; k component latent mixture gaussian negative binomial model in a finite structure	w/ two conditions	longitudinally measured two series time course; at least two replicates and three time points per condition are needed; balanced design	T(4,6,10), R(3,6), C(2)	DESeq2, splineTimeR, Next maSigPro-GLM, ImpluseDE
TimeMeter [64]	dynamic time warping algorithm; metrices for similar temporal patterns; progression advance scores	w/o conditions	comparative method for two single-series time course data; no parameters for dispersion and biological replicates	T(9,16,26), R(NA), C(NA)	
PairGP [61]	non-stationary Gaussain process; exponentiated quadratic kernel;	w/ two ore more conditions	longitudinal time course with paired multi-group conditions	T(9),R(3), C(2,3,4)	base model w/o pairing effect
GPrank [81]	Gaussian process; radial basis kernel; logarithm of Bayes Factor for two models	w/o condition	balanced/unbalanced single-series time course	T(10), R(0–3), C(NA)	
Dream [60]	linear mixed model; limma/voom-incorporated Bioconductor package; multiple random effects: Kenward-Roger approximation for small samples;	w/ two or more conditions	longitudinally meausred multi-series of time course data	R(2–4) Individuals(4–50)	DESeq2, limm/voom, macau2
rmRNAseq [82]	genelized linear model; voom-incorporated R package; continuous autoregressive correlation; parametric bootstrap; residual maximum likelihood;	w/ two or more conditions	longitudinally measured multi-series of time course data	T(4), R(4), C(2)	edgeR, DESeq2, splineTimeR, ImpulseDE2
Comparative study [36]	comparison of dynamic gene-wise testing tools	data sets w/ two conditions	next maSigpro, DyNB, EBSeq-HMM, ngsp, lmms, splineTimeR, ImpulseDE2,	T( >=4), R(3), C(2)	

In the # of replicates, it represents the number of biological replicates in each of studies. In the data type, one represents a single(one)-series of time course data without conditions at each time point, two represents two-series of time course data, i.e., a case-control type of time course data containing two levels of the condition group at each time point, and multi-represents the data type where there are at least three condition levels at each time point.

**Table 2 genes-12-00352-t002:** Dynamic gene-by-gene (gene-wise) testing tools for periodical time course RNA-Seq data.

Dynamic Tools in Periodicity	Method	Exeternal Factors at a Time	Experimental Design	Competiting Methods
JTK_CYCLE [74]	Jonckheere_Terpstra Kendal’s statistics	w/o condition	single series periodical time course	COSOPT, Fisher’s G test
MetaCycle [17]	meta tool among ARSER, JTK_CYCLE, LS	w/o condition	single series periodical time course	
RAIN [102]	umbrella alternatives for steep rise and slow falling, or vice versa	w/o condition	single series periodical time course	JTK_CYCLE
DODR [103]	parametric and non-parametric non-gaussian measurement for noise	w/ two conditions	two series periodical time course	
LimoRhyde [101]	cosinor regression	w/ two or more conditions	two or multi-series time course	DODR

**Table 3 genes-12-00352-t003:** State-of-art batch detection tools.

Tools	Features
ARSyN [113]	microarray dynamic time course data; batch-free expression data after removal of unwanted biases; both known and unknown batch factors; multiple batch factors
Combat-Seq/Combat [55]	microarray and RNA-Seq static and dynamic time course data; known batch factors; batch-free data after removal of unwanted biases; multiple batch sources
RUV-Seq [46]	RNA-Seq static data; both known and unknown batch factors; estimates of batch effects; user-defined k value for hidden factors
svaseq/sva [52,91]	microarray and RNA-Seq static data; both known and unknown batch factors: estimates of batch effects
gPCA [90]	microarray and RNA-Seq static data with one known batch factor;
Harman [58]	microarray and RNA-Seq static and dynamic data w/ one known batch factor
MMD-ResNet [89]	static and dynamic RNA-Seq or other types of omics data w/ both known and unknown batch source

**Table 4 genes-12-00352-t004:** Coherent gene-to-gene dynamic methods for non-periodical time course data.

Coherent Gene-to-Gene Dynamic Methods	Method	Exeternal Factors at a Time	Experimental Design	Competiting Methods
Tcgsaseq [8,11]	variance component score; linear mixed effects model; permutation test	w/o condition	single series of longitudinally measured time course	voom-Roast, Roast, edgeR-Roast
FunPat [23]	best fitted model selection for each gene; linear model-based clustering; integration with the given gene ontology and pathway information	w/o condition	single- or multi-series of time course	edgR, maSigPro, FPCA; HC, k-means, MBC
DPGP [116]	inferene of time-varying trajectories; Dirichlet prior mixture models; Gaussian process	w/o condition	single series of time course	HC, k-means, Mclust, SplineCluster, GIMM, BHC
LPWC [117]	lag-penalized weighted correlation; Gaussian penalty score;	w/ two conditions	single-series of time course	HC, k-means, DTW w/ HC, STS w/ HC

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
