# Peer review of "Temporal Dynamic Methods for Bulk RNA-Seq Time Series Data"

_genes, 2021, doi:10.3390/genes12030352_

Round 1

Reviewer 1 Report

This is a very well written and comprehensive manuscript. I have no additional methods nor major corrections to add to the review.

Minor Corrections:

-Table 1 has a few issues: 1) the legend refers to the "Data Type" being 1, 2, or 3 but the Data Type column has text in it; 2) the title for parameter setting needs an "R" added so that "T(time), (replicate), C(condition)" turns into "T(time), R(replicate), C(condition)".

-I suggest breaking up table 1 into multiple tables so that each section of the table is more readable.

-The sub-headings in "1. Single gene-by-gene testing for non-periodical time course data" are intermittently bold, italic, change font sizes, space before ref only for some refs, etc. -- this problem continues throughout the other sections as well.

Author Response

Dear Reviewers and Editors of Genes:

V.S.O and R.W.L, authors, sincerely thank you for the valuable comments from reviewers that have not been thoroughly discussed in the previously submitted files and should be addressed carefully further in this revision.

We sincerely accepted and addressed all of comments in this revision and please take a closer look at the highlighted main text and tables accordingly, based on the point-by-point responses on our revision.

Reviewer # 1:

This is a very well written and comprehensive manuscript. I have no additional methods nor major corrections to add to the review.

Minor Corrections:

-Table 1 has a few issues: 1) the legend refers to the "Data Type" (Experimental designs) being 1, 2, or 3 but the Data Type column has text in it; 2) the title for parameter setting needs an "R" added so that "T(time), (replicate), C(condition)" turns into "T(time), R(replicate), C(condition)".

==> Thank you for the valuable comment.

We represnet the various different types of experimental designs (Data types) for each dynamic method instead of rerpesenting simple numberings (1 to 3) in Table 1.

And we corrected R (replicate) in Table 1 in the current version.

-I suggest breaking up table 1 into multiple tables so that each section of the table is more readable. (non-periodical data for Table and periodical methods for Table)

==> Thank you for the valuable comment.

As suggetsed, we split the Table 1 with Table 2 for periodical time course data.

-The sub-headings in "1. Single gene-by-gene testing for non-periodical time course data" are intermittently bold, italic, change font sizes, space before ref only for some 

==> Thank you for the comment for the sub-heading format. 

Yes, we corrected.

Thanks to those insightful comments, we were able to reconsider our manuscript very carefully in different angles.

Thank you for your consideration. I look forward to hearing from you.

Sincerely yours,

Vera-Khlara S. Oh, PhD

United States Department of Agriculture,

Agricultural Research Service,

Animal Genomics and Improvement Laboratory,

Beltsville, MD, USA, 20705

Department of Computer Science and Statistics,

College of Natural Sciences,

Jeju National University,

Jeju City, Jeju Do, S. Korea, 

Phone: +1-301-504-5185

Fax: +1-301-504-8414

Email: sshshoh1105@gmail.com

Author Response

RESPONSE LETTER

February, 14th, 2021

Dear Reviewers and Editors of Genes:

V.S.O and R.W.L, authors, sincerely thank you for the valuable comments from reviewers that have not been thoroughly discussed in the previously submitted files and should be addressed carefully further in this revision.

We sincerely accepted and addressed all of comments in this revision and please take a closer look at the highlighted main text and tables accordingly, based on the point-by-point responses on our revision.

Reviewer # 2:

  1. a: Modify the title to make it obvious
  2. b: Modify the abstract specifically at:

== > Thank you for the valuable comment.

As  suggested, we modifed the titel and abstract in the current version.

  1. c: Please add an introductory paragraph that:

==> Thank you for the insightful comment.

As suggested, we added the introductory paragraph in the beginning of section 2. for the RNA-Seq data (Wang et al., (2009), Li & Li (2019)), static versus dynamic methods briefly and accordingly, we added the corresponding references.

2. missing references:

==> Thank you for the missing references for the more recently updated methods.

As recommended, we added the missing references except PsedotimeDE as this dynamic method is for scRNA-seq data. 

For b. c method, we have already referred to as the literatures to be ciated in the previous version of manuscript on the initial submission.

3. Table 1 should be split with Table 2.

==> Thank you for the comment.

As suggested, we corrected.

4. Table 2 has been mentioned in main text.

==> Thank you for the valuable comment. Yes, we added the Table 2 in main text. (in 279 line)

5. Section 4 needs a Table for summary.

==> Thank you for the comment.

As suggested, we addred Table 4 in section 4.

6. Thank you for the valuable comment.

For the static gene-to-gene methods, especially the majority of static unsupervised clustering methods are based on simple correlation or distance metrics to measure similarity of gene expression patterns between condition groups and intuitively, the shuffling of the labels of samples does not affect the results and obtain the identical results.

However, the dynamic clustering or gene set tests have been implemented to account for time dependent structures in which the observed samples are measured in ordered time points such as the longitudinal time course data, resulting in the different clusters or gene sets when switching the sample labels.

We referred to as the previous literatures for this paragraph.

  1. Thank you for the comment. As mentioned, the year on the citation has been corrected.

Thanks to those insightful comments, we were able to reconsider our manuscript very carefully in different angles.

Thank you for your consideration. I look forward to hearing from you.

Sincerely yours,

Vera-Khlara S. Oh, PhD

United States Department of Agriculture,

Agricultural Research Service,

Animal Genomics and Improvement Laboratory,

Beltsville, MD, USA, 20705

Department of Computer Science and Statistics,

College of Natural Sciences,

Jeju National University,

Jeju City, Jeju Do, S. Korea, 

Phone: +1-301-504-5185

Fax: +1-301-504-8414

Email: sshshoh1105@gmail.com

Thanks to those insightful comments, we were able to reconsider our manuscript very carefully in different angles.

Thank you for your consideration. I look forward to hearing from you.

Sincerely yours,

Vera-Khlara S. Oh, PhD

United States Department of Agriculture,

Agricultural Research Service,

Animal Genomics and Improvement Laboratory,

Beltsville, MD, USA, 20705

Department of Computer Science and Statistics,

College of Natural Sciences,

Jeju National University,

Jeju City, Jeju Do, S. Korea, 

Phone: +1-301-504-5185

Fax: +1-301-504-8414

Email: sshshoh1105@gmail.com

Reviewer 3 Report

See below.

Author Response

None

Round 2

Reviewer 2 Report

Thank you to the authors for their response and the changes made.  My last comment is there seems to be a reference to figures that are missing from the paper draft.  See below.

“Therefore, we recommend the development of a 274 complete analytical pipeline to better characterize dynamic changes and to reduce mis-275 leading results (Figure 1-(1) and Figure 1-(2) with detailed descriptions for each step).”

Author Response

 February, 19th, 2021

Dear Reviewers and Editors of Genes:

V.S.O and R.W.L, authors, sincerely thank you for the thoughtful comment regarding the missing references for Figure 1-(1) and Figure 1-(2).

As suggested, we incorporated the references in the main text. 

Thank you for your consideration. I look forward to hearing from you very soon!

Sincerely yours,

Vera-Khlara S. Oh, PhD

United States Department of Agriculture,

Agricultural Research Service,

Animal Genomics and Improvement Laboratory,

Beltsville, MD, USA, 20705

Department of Computer Science and Statistics,

College of Natural Sciences,

Jeju National University,

Jeju City, Jeju Do, S. Korea, 

Phone: +1-301-504-5185

Fax: +1-301-504-8414

Email: sshshoh1105@gmail.com